# An Unusual Case of Collision Testicular Tumor in a Female DSD Dog

**DOI:** 10.3390/vetsci10040251

**Published:** 2023-03-27

**Authors:** Claudia Rifici, Emanuele D’Anza, Viola Zappone, Sara Albarella, Valeria Grieco, Marco Quartuccio, Santo Cristarella, Cornelia Mannarino, Francesca Ciotola, Giuseppe Mazzullo

**Affiliations:** 1Department of Veterinary Sciences, University of Messina, via G. Palatucci, 98168 Messina, Italy; 2Department of Veterinary Medicine and Animal Production, University of Naples Federico II, via Delfino 1, 80137 Naples, Italy; 3Department of Veterinary Medicine and Animal Science, University of Milan, via dell’ Università 6, 26900 Lodi, Italy

**Keywords:** collision tumor, dog, testicular tumor, disorders of sexual development

## Abstract

**Simple Summary:**

Collision tumors (CT) consist of two independent neoplasm populations. Disorders of sexual development (DSDs) are characterized by various abnormalities of the genital tract. A phenotypically female 8-year-old Jack Russell terrier was submitted for clinical evaluation due to anomalous vaginal discharge and non-pruritic cutaneous bilateral symmetrical alopecia on the flanks. A voluminous mass was detected in the left quadrant area of the abdominal region and confirmed by ultrasound and necropsy examination. In the abdominal cavity, the left gonad was increased in size while the right one was decreased, as was the uterus. Histologically, the left gonad revealed a double neoplastic component (sustentacular tumor and interstitial cell tumor), whereas the right gonad showed coarctated seminiferous tubules. PCR amplification of the *SRY* and *AMELX* genes revealed the absence of a Y chromosome. To the best of our knowledge, this is the first report describing a case of a coexisting testicular collision tumor in a dog with testicular SRY-negative DSD.

**Abstract:**

Collision tumors (CT) consist of two independent neoplasms with distinct neoplastic populations. Disorders of sexual development (DSDs) are characterized by atypical sexual development leading to various abnormalities of the genital tract. Sex reversal (SR) syndromes are a type of DSD characterized by a discrepancy between chromosomal sex and gonadal development (testes/ovaries) and the presence or the absence of the *SRY* gene. A phenotypically female 8-year-old Jack Russell terrier dog was referred due to anomalous vaginal discharge and non-pruritic cutaneous bilateral symmetrical alopecia on the flanks. During abdominal palpation, a voluminous mass was detected in the left quadrant area, later confirmed by ultrasound. The owner decided to proceed with euthanasia and necropsy. In the abdominal cavity, the left gonad was increased in size, the right one and the uterus were decreased, and the vagina and vulva appeared to be thickened. Histologically, both gonads were revealed to be testes: the left one was affected by a double neoplastic component (sustentacular tumor and interstitial cell tumor), whereas the right gonad showed coarctated seminiferous tubules. PCR amplification of the genes *SRY* and *AMELX* revealed the absence of the MSY region of the Y chromosome. To the authors’ knowledge, this is the first report describing a case of a testicular collision tumor in a DSD SRY-negative dog.

## 1. Introduction

Collision tumors (CT) are clinically represented by a single tumor, but are histologically characterized by two independent neoplasms, each originating from two morphologically distinct neoplastic cell type populations [1,2,3,4]. Disorders of sex development in dogs with female karyotypes (XX DSD) are quite common. The most common canine disorder of sex development (DSD) manifests as testes or ovotestes without gametogenic activity, a normal female karyotype (78, XX), and a lack of the SRY gene. This abnormality, termed testicular or ovotesticular XX DSD, is also quite common in other mammals, including humans and livestock species (goat, pig, horse, etc.) [5]. Disorders of sexual development (DSDs) are characterized by discrepancies of the chromosomal, gonadal, and anatomical sex, and can occur at any stage of sexual development, leading to various abnormalities of the genital tract. The pathway of sexual development is very complex, and involves a large number of genetic and hormonal factors. The correct classification of DSDs, although indispensable for the purpose of identifying the causes, has always been the subject of discussion, and is mainly based on the chromosome complement (normal XX, normal XY, or abnormal). Particularly confusing is the use of terms such as sex reversal, pseudohermaphroditism, and true hermaphrodite. For this reason, attempts are currently being carried out, thanks to modern diagnostic aids, to confirm the classification of each individual case with clinical diagnostic findings [6]. Many studies have reported causative mutations or linked genetic markers in dogs using various molecular methods, such as cytogenetic mapping (fluorescence in situ hybridization, FISH) and multiplex ligation-dependent probe amplification (MLPA) [7]. Thus, the different forms of DSDs are classified according to the type of gonads (testis, ovary, or ovotestis), chromosomal arrangement (XX, XY, XX/XY, X0, XXX, XXY, etc.), and presence of the *SRY* gene (positive or negative). From a genetic point of view, the presence or absence of the *SRY* gene and its normal functioning represent a key element in the distinction of DSDs and in the subsequent search for the cause, since expression of the SRY gene triggers the development of undifferentiated gonads into testicles. In the absence of the *SRY* gene, there is a greater expression of genes responsible for the formation of the ovaries (*beta-catenins*, *RSPO1*, *WNT4*, and *FOXL2*) and for the development of a female reproductive systems [8,9,10]. The coexistence of DSDs and neoplastic processes has rarely been reported in canine species [6,9,11,12]. In this case report, we describe the pathological findings of a testicular collision tumor detected in a female dog with disorders of sexual development.

## 2. Materials and Methods

An 8-year-old, phenotypically female Jack Russell terrier was referred to a gynecological specialist because of anomalous vaginal discharge. On admission, the dog was in good body condition (BCS = 3/5) and had normal skeletal and muscular development, normal mucous membranes, no sensory impairment, and normal palpable lymph nodes, but had non-pruritic cutaneous bilateral symmetrical alopecia on the flanks. However, the anovulvar distance was normal and the vulva was mildly edematous. The vagina, clitoris, and urethral orifice aligned with normal anatomical ratios. During abdominal palpation, a voluminous mass was detected in the left quadrant area. Upon ultrasound examination, it was possible to observe an encapsulated mass (5 × 4 cm in size) with a presence of hyperechoic mixed with hypoechoic parenchyma areas (Figure 1a). This appearance, typical of complex masses, led to a suspicion of neoplasia. Computerized tomography (CT) and explorative laparotomy were proposed but declined by the owner. After two months, the bitch was taken to the emergency room of the Veterinary Hospital of the Department of Veterinary Sciences, University of Messina (Italy), because of a worsening of general health conditions, and a blood transfusion was necessary due to the anemia and mild jaundice. Hematological analysis indicated anemia, leukocytosis, and thrombocytopenia (hematocrit: 16.7%, hemoglobin: 6.2 g/dL, RBC: 2.87 M/μL, WBC: 0.61 K/μL, platelets: 0 K/μL). Moreover, serological investigations were carried out, with negative results for *Leishmania* spp, *Ehrlichia canis*, *Borrelia burgdorferi*, *Anaplasma*, *Dirofilaria immitis*, and *Angiostrongylus vasorum*. After a further deterioration of clinical conditions, the owner declined any treatment and decided to proceed with euthanasia and necropsy.

During the necroscopy, samples from all the genital systems were collected, fixed in 10% buffered formalin, routinely processed for histology, and paraffin-embedded; then, 3–4 μm thick sections were obtained and stained with hematoxylin and eosin (HE).

DNA was extracted from paraffin-embedded tissue (FFPE) (heart and mass of abdominal cavity) with ReliaPrep™ FFPE gDNA Miniprep System (Promega, Madison, WI, USA).

PCR analyses were carried out to check for the presence of the *SRY* and *AMELX/Y* genes [13].

## 3. Results

Necropsy revealed the presence of a mass at the site of the left gonad, while the right gonad was decreased in size, as was the uterus. On the contrary, the vagina and vulva appeared thickened. (Figure 1b). Histologically, in the uterus, the typical three-layer structure was absent. The perimetrium was replaced with fibro-connective tissue, while both the myometrium and endometrium were strongly reduced and poorly vascularized (Figure 1c). The firm structure corresponding to the right gonad was constituted by fibro-connective tissue in both the testes and the epididymis. Only in the testes was it possible to observe the occasional presence of coarctated pseudo-seminiferous tubules, surrounded by a wavy, adherent, and thickened basement membrane. Both germ cells and sustentacular (Sertoli) cells were absent (Figure 1d). In detail, the 5 × 4 × 3 cm mass in correspondence with the left gonad was firm-elastic in consistency and showed an external smooth surface with a dense network of vascular structures. On the cut section, the mass showed a variegated appearance, ranging from pale to dark red, and was organized in pseudo-lobular patterns (Figure 2). Histology revealed that the left gonad’s mass was consistent with that of a testicle totally substituted by a tumor with double neoplastic components. One component was characterized by oval to polyhedral cells, which were arranged in lobules separated by a fibrovascular stroma. The neoplastic cells revealed a variable quantity of granular or vacuolated eosinophilic cytoplasm and round euchromatic nuclei. Mitotic figures were rare. Large areas of necrosis were also present. These findings were consistent with a diagnosis of interstitial cell tumor (Figure 2a). The other component was characterized by the presence of pseudo-tubular structures lined by spindle-shaped or rounded cells which were arranged perpendicularly to the basement membrane. Neoplastic cells had a moderate amount of faintly eosinophilic cytoplasm, with large, empty vacuoles and oval to elongated nuclei. Mitotic figures were rare. These findings led to a diagnosis of a sustentacular tumor (previously called a Sertoli cell tumor) [14] (Figure 2b).

To establish the genetic sex of the dog, the remaining paraffin tissue was used to extract DNA (ReliaPrep™ FFPE gDNA Miniprep System (Promega, Madison, WI, USA). Initially, the protocol indicated by the manufacturer was followed, but given the poor results, small changes were made to the protocol in order to improve the purity of the sample. In particular, 2 20 mg pieces of each block of paraffin (heart and mass) were cut, moved to a 1.5 mL tube, and treated for deparaffinization according to the xylene protocol indicated in the kit user manual, which was performed twice. After complete ethanol evaporation, 200 μL of lysis buffer and 20 μL of proteinase K (20 mg/mL) were added to each sample. All of the samples were incubated overnight at 56°. The extraction was then completed following the instructions in the user manual, and elution was performed in 30 μL of elution buffer. All of the extracted DNA was tested by PCR. A first PCR, using primers for dog microsatellite INU055, was performed to check whether the genomic DNA was amplified. Then, extracted DNA was analyzed for *SRY* and *AMELX/Y* detection (see Table 1 for PCR details). Negative (mix without DNA and mix with female dog DNA) and positive controls (mix with male dog DNA) were added to all PCR tests. No amplifications were observed for the *SRY* or *AMELY* genes in the case study, while the expected fragments for *AMELX* and microsatellite (INU055) were positively amplified (Figure 3). According to these results, the dog could be classified as a female dog with a testicular SRY-negative DSD.

## 4. Discussion

This report describes a case of canine DSD with the coexistence of a double testicular neoplasia. The findings revealed a case of a phenotypically female dog with a normal vulva and vagina, hypoplastic bicornuate uterus, and male gonads. These were represented by an atrophic right testis and a neoplastic left testis. DNA analysis highlighted the absence of some typical regions of the Y chromosome, and, in particular, of the *SRY* gene, the main actor in the embryonic development of the testicles. Since it was not possible to carry out a study on the karyotype of this case, although the genetic data obtained by PCR show the absence of typical regions of the Y chromosome, it is not possible to establish the chromosomal arrangement with certainty, nor, therefore, to exclude, for example, a deletion of a very large region of the Y chromosome. As a matter of fact, the most common form of DSD in dogs is the XX (SRY-negative) DSD, which is characterized by variable virilization of the external genitalia and the presence of a uterus, ductus deferens, and testes or ovotestes. Although the observed phenotype in this case is very typical of dogs with XX SRY-negative DSD [15,16], without a karyotype analysis or a complete genome assembly, the data restricted the classification of the case to that of a dog with testicular SRY-negative DSD.

Several types of DSDs have been described based on failures that can lead to alterations in the normal development of the genital apparatus. Several types of DSDs are related to altered functioning of the XY male karyotype containing the SRY gene, which is located on the Y chromosome and is responsible for the formation of male gonads (testes) and an internal and external male phenotype. DSDs can also be associated with altered functioning of the XX zygotes, which typically develop female gonads, the ovaries, and an internal and external female phenotype [17]. As evidenced by the observed pathological phenotypes, in XX SRY-negative DSDs, despite the absence of the SRY gene, testes develop from primordial gonads. In dogs, this disorder occurs with a frequency that provides sufficient information to develop an excellent animal model for studying this developmental anomaly [15].

Not enough studies have been performed to evaluate the incidence of intersex conditions in dogs. The coexistence of DSDs and neoplastic processes is quite rare in the veterinary literature, although it is not always easy to identify the onset of a neoplasm in reproductive organs that are not fully (normally) developed [6,9,11]. In the literature, only a few cases of disorders of development associated with testicular neoplasia have been reported. Kelly et al. (1976) described a case of a phenotypically male dog characterized by a hypoplastic uterus and testes with neoplasia [18]. Dizmira et al. (2015) reported a case of a bitch with cryptorchid testes, one of which showed the presence of a gonadoblastoma [12]. Herndon et al. (2012) described a phenotypically female dog with testicles retained in the groin region, one of which was affected by a Sertoli cell tumor [11]. Moreover, different cases of persistence of the Mullerian duct and testicular tumors, have been described [19,20], and recently, Schwartz et al. reported a phenotypically female DSD dog with XX/XY leukocyte chimerism and testicular mixed tumors [21]. Only few cases of DSD with neoplasms have been described in dogs, and this is probably due to premature castration, which reduces the probability of tumor development. In the present report, a dog with testicular *SRY*-negative DSD showed a double tumor in the altered left gonad: a sustentacular tumor and an interstitial cell tumor. Although it is not easy to identify the onset of a neoplasm in reproductive organs that are not fully developed or abnormal [12], in the present case, some clinical signs could be attributable to feminizing paraneoplastic syndrome due to a sustentacular tumor, such as the bilateral symmetrical alopecia, edematous and thickened vulva, and probable estrogen myelotoxicosis with bone marrow hypoplasia, which is clinically characterized by anemia [22,23]. Moreover, it would seem that there is a significantly higher risk of neoplasia in subjects with undescended testes than in those with eutopic testes [24,25]. In dogs, the testicles are elective sites for collision tumor detection, and the most frequent combination is represented by seminoma and sustentacular tumors [26,27,28]. In this report, we describe an unusual collision testicular tumor composed of an interstitial cell tumor and a sustentacular tumor in a female dog with male gonads. Recently, Rifici et al. (2021) described the presence of testicular collision, sustentacular, and Leydig cell tumors with the concomitant presence of seminoma scrotal melanoma, and cutaneous mast cell tumors in the contralateral testicles of dogs [29]. The developing mechanisms of CTs are unknown, but the origin is attributed to various etiopathogenetic hypotheses: random development in the same venue of two different primary tumors; simultaneous development under the influence of the same carcinogenic factor (i.e., radiation) of two morphologically distinct neoplasms; and alteration of the microenvironment induced by the presence of a primary tumor which, in turn, allows for the development of a secondary tumor [30,31]. Although they are rare conditions, in dogs, collision tumors have been described mostly to be associated with melanoma [1,2], hemangiosarcoma [32,33], and mast cell tumors [29,33,34,35].

## 5. Conclusions

The relation between DSD and neoplastic processes is common in humans [36], but there is little data relating to dogs. It is, therefore, essential to investigate this pathological association in the canine species as well. Unlike the murine model, dogs do not develop artificially induced, but instead spontaneous, pathologies, representing an important experimental model for humans, with whom dogs share many pathologies. To the best of our knowledge, this is the first report describing a case of a coexisting testicular collision tumor in a DSD SRY-negative dog. This report describes a finding of exceptional rarity, both due to the lack of a clear clinical pathological signal and to the presence of a collision tumor.

## Figures and Tables

**Figure 1 vetsci-10-00251-f001:**
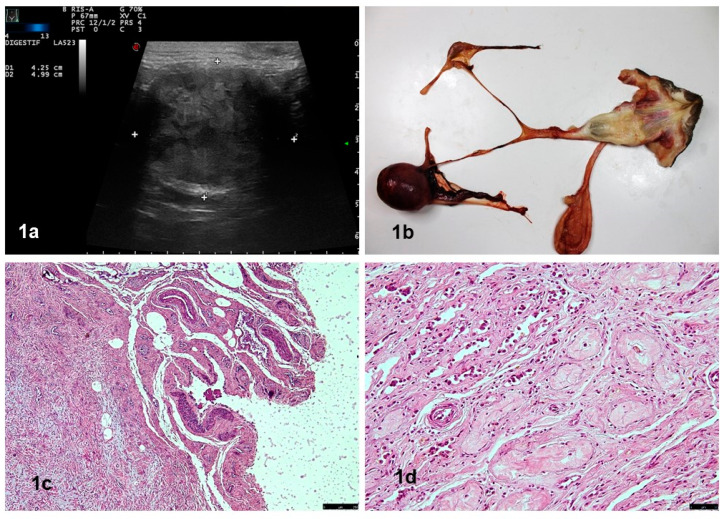
(**a**) Encapsulated mass (5 × 4 cm in size) with a presence of hyperechoic mixed with hypoechoic parenchyma areas. (**b**) Uterus was strongly hypoplastic, with a small body, two slender uterine horns, and a thickened vagina and vulva. (**c**) Uterus: the typical three-layer organization was absent. HE, 5×; (**d**) right gonad: fibro-connective tissue with the occasional presence of coarctated pseudo-seminiferous tubules. HE, 20×.

**Figure 2 vetsci-10-00251-f002:**
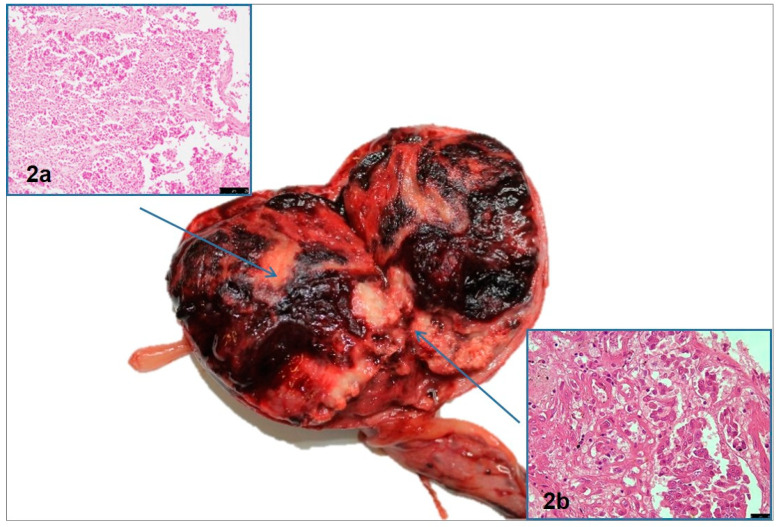
The left gonad mass (5 × 4 × 3 cm), On the cut section, a variegated appearance can be observed, characterized by pseudo-lobular patterns with double neoplastic components. (**a**) Interstitial cell tumor, HE (20×), scale bar: 25 μm; (**b**) sustentacular tumor, HE (40×), scale bar: 25 μm.

**Figure 3 vetsci-10-00251-f003:**
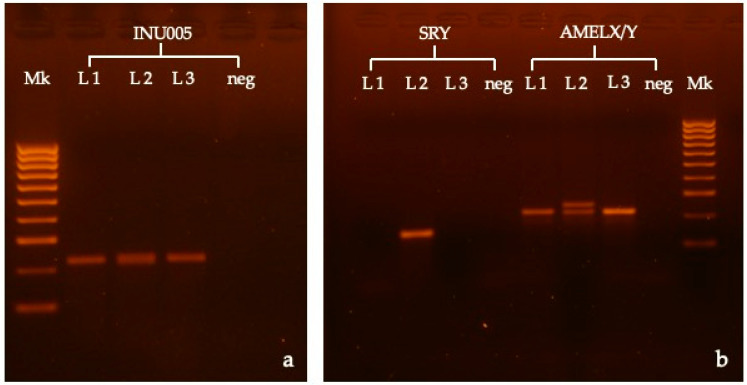
Agarose gel (1.8%) electrophoresis of PCR products of INU005 (**a**), SRY, and AMELX/Y (**b**). Mk = 100 bp molecular marker (HyperLadder 100 bp, BIO-33056, Bioline, London, UK), L1 = case study, L2 = male dog control, L3 = female dog control, neg = negative control. Please find the original images in Appendix A.

**Table 1 vetsci-10-00251-t001:** Primers and PCR conditions.

Gene	Primers	Sequence (5′-3′)	Fragment Size (bp)	T° Annealing	Genome Pos
*INU055*	*U*	CCAGGCGTCCCTATCCATCT	*210*	60°	Chr10: 67,428,683–67,428,473
	*L*	GCACCACTTTGGGCTCCTTC			
*Sry*	*CFA-SRY-F2*	GCAGGTGCACGTAGATGAGA	*142*	57°	ChrY: 1,350,170–1,350,312
	*CFA_SRY_Short_R3*	TGTGGTACTCCTGTTGCAG			
*Amelx/y* [11]	*CFA_AmelxF*	ATAATGACAAAGAAAACATGAC	*215/247*	55°	ChrX: 7,828,350–7,828,136
	*CFA_AmelxR*	CTGCTGAGCTGGCACCAT			

PCR was performed in a 30 μL reaction system containing 60 ng genomic DNA, dNTPs (200 μM each dNTP, dNTP Mix, Promega, Madison, WI, USA), 0.75 U Taq DNA polymerase (GoTaq G2 DNA Polimerase, Promega, Madison, WI, USA), and 0.1 μM of each primer. The PCR condition were 95 °C for 5 min; 35 cycles of 94 °C for 30″, with the annealing temperature according to the primer set for 30″, 72 °C for 20″; and a final extension at 72 °C for 10′.

## Data Availability

The data presented in this study are available upon request from the corresponding author.

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
