# Peer review of "An Unusual Case of Collision Testicular Tumor in a Female DSD Dog"

_vetsci, 2023, doi:10.3390/vetsci10040251_

Round 1

Reviewer 1 Report

the document is well written, original and represents a milestone for the future. well done

Reviewer 2 Report

The manuscript present an interesting and well documented case. It is in a scope of the Veterinary Sciences journal. I suggest to consider several minor remarks (see below)

Title. Should be rewritten, e.g. “An unusual case of collision testicular tumor in a female DSD dog”

Line 56-58. It should be mentioned that the classification is based on sex chromosome complement (abnormal, normal XX or normal XY)

Line 70. Should be re-written, since expression of the SRY triggers development of undifferentiated gonad into testicle

Line 74.  FOXL2 should be also mentioned

Line 79. More information concerning external genitalia should be added (if these information are available): anus-vulva (normal?), size of clitoris (normal, enlarged?), position of urethral orifice?

Lines 108-109. Was the epididymis present?

Line 203. Such PMDS case was also reported by Dzimira et al. (2018, Journal of Comparative Pathology 161: 20-24)

219. should be “with”

Reviewer 3 Report

dear authors , thank you for your work, which describes in detail a case of a disorder of sexual development associated with a collision tumor. I find the case very interesting and well written, I would just like you to clarify two small details.The animal had non-pruritic cutaneous bilateral symmetrical alopecia on the flanks. The most likely diagnosis would be endocrine dermatosis. No skin biopsy was performed?

The histopathology of the tumor is not perceptible. Is it possible to provide higher magnification on photographs 2a and 2b?

Thank you again for your work

Reviewer 4 Report

Comments to the Author

General comments: 

This manuscript describes a rare case report of testicular collision tumor in a dog with SRY-negative disorders of sexual development (DSDs). The Authors pathologically investigated genital organs, and determined as a testicular SRY negative DSD by genetic analysis. Recently, dogs have received attention as an ideal model animal, such as in cancer research. As mentioned in the manuscript, this case study provides important insights for both human and veterinary medicine.

Specific comments: 

1)     Page 1 line 17: revised word (dog) remains unerased.

2)     Page 2 line 88: computerized tomography (TC) => computerized tomography (CT)

3)     Page 4 figure 2: Please add a scale bar to figure 2a. And I recommend to indicate the scale bar length in figure legends.

4)     Page 5 figure 3b: Please clearly indicate which lanes are SRY and AMELX/Y.

5)     Page 6 line 219: ~ and Leydig cell tumors, whit the concomitant ~. Is this a misspelling of “what”?

6)     Can the authors provide any clinical information of the period between the first visitation and the emergency visit after two months?

7)     One of the limitations of this study is the lack of karyotype analysis. I personally recommend to add some literature review about karyotype analysis.

8)     Recently, Schwartz et al. reported phenotypically female DSD dog with XX/XY leukocyte chimerism and testicular mixed tumors (https://pubmed.ncbi.nlm.nih.gov/35898552/). Please consider citing this report.
